

# Reweighting Lefschetz Thimbles

Stefan Blücher[1], Jan M. Pawlowski[1,2], Manuel Scherzer[1], Mike Schlosser[1],
Ion-Olimpiu Stamatescu[1], Sebastian Syrkowski[1] and Felix P.G. Ziegler[1]

**1** Institut für Theoretische Physik, Universität Heidelberg,
Philosophenweg 16, 69120 Heidelberg, Germany
**2** ExtreMe Matter Institute EMMI, GSI, Planckstraße 1, D-64291 Darmstadt, Germany

## Abstract

One of the main challenges in simulations on Lefschetz thimbles is the computation of the relative weights of contributing thimbles. In this paper we propose a solution to that problem by means of computing those weights using a reweighting procedure. Besides we present recipes for finding parametrizations of thimbles and anti-thimbles for a given theory. Moreover, we study some approaches to combine the Lefschetz thimble method with the Complex Langevin evolution. Our numerical investigations are carried out by using toy models among which we consider a one-site $z^4$ model as well as a $U(1)$ one-link model.



# 1  Introduction

QCD at vanishing and finite temperature is one of the best tested theories in high energy physics. At the present moment theoretical predictions from first principle lattice simulations match remarkably well with experimental data for instance from heavy ion collisions, see e.g. [1–3]. However, at finite chemical potential lattice simulations suffer from the sign problem, which a priori prohibits simulations based on importance sampling. This severely limits the access to the largest part of the QCD phase diagram. By now, there are many approaches towards a solution of the sign problem. Overviews addressing developments in finite density QCD over the last years can e.g. be found in [4–8]. Amongst those are Taylor expansions [9], simulations at imaginary chemical potential [10, 11], reweighting [12], the density of states method [13], dual formulations [7], the Complex Langevin method [14, 15] and the Lefschetz thimble method [16, 17]. So far none of those methods have been able to give reliable results for $\mu/T \gtrsim 1$. In this work we focus on the Lefschetz thimble approach. In Euclidean space-time the Lefschetz thimble approach has been applied to bosonic theories as well as to (low-dimensional) QCD in [17–22]. Recent applications to fermionic theories (such as the Thirring model) can be found in [23, 24]. Moreover, field theories in Minkowski space-time formulated on the Schwinger-Keldysh contour have been studied using the thimble formalism in [25]. Algorithmic improvements to the holomorphic gradient flow method were proposed in [26]. Recent contributions in the field more generally involve complex manifolds close to Lefschetz thimbles that are optimized such that they ameliorate the sign problem, see e.g. [27–30].

As we see later, the CLE and the Lefschfetz thimbles are closely related. Studies investigating the interplay and the connection between the two approaches can be found in [31, 32].

The Lefschetz thimble method relies on a deformation of the integration path. By construction the imaginary part of the action is constant on the transformed paths, which are called Lefschetz thimbles. There are two basic algorithmic frameworks [17, 23] providing recipes for Monte Carlo simulations on the Lefschetz thimbles. The first employs Monte Carlo simulations directly on the thimbles. The latter continuously deforms the original integration path close to the actual thimbles to lessen the sign problem.

In this work we address a few key challenges to the the Lefschetz thimble method and propose algorithmic improvements. One of the main problems with Monte Carlo simulations on Lefschetz thimbles is to determine the weights of the thimbles relative to each other. This difficulty arises as the original path integral is decomposed into a sum of integrals over multiple thimbles. We show that this difficulty can be overcome by a standard Monte Carlo determination of the ratios of the real partition functions on the thimbles. This is facilitated by a novel reweighting procedure which is generally applicable to field theories. In this work we assume prior knowledge on a parametrization of the contributing thimbles. To find this

parametrization we propose two algorithms which can be generalized to higher dimensional theories. However, the reweighting procedure does not rely on knowing a parametrization. Our ideas are put to work in simple models, i.e. ordinary integrals. Among those we consider a one-site quartic model with a $\frac{\lambda}{4}z^4$ term as well as a $U(1)$ one-link model.

The paper is organized as follows. We start by briefly revisiting the idea behind Lefschetz thimbles, see Sec. 2. In Sec. 3 we propose two algorithms to find thimbles and their parametrizations necessary for Monte Carlo integration. In Sec. 4 we present our idea of sampling on multiple thimbles taking into account the relative weights of different thimbles. Sec. 5 introduces the toy models we use for numerical investigations together with our results. We conclude this paper in Sec. 6. During the research for this paper we have also developed many ideas to combine the Complex Langevin evolution and the Lefschetz Thimble method. While none of those approaches lead to generally applicable algorithms, they still provide some useful insight into the structure of the models, hence we give some of those ideas and corresponding results in App. C.

## 2 The Lefschetz thimble method

In the following we briefly revisit the Lefschetz thimble method. The idea behind the approach is to rewrite the path integral measure over a real manifold [16] to circumvent the sign problem by allowing Monte Carlo sampling on this manifold [17]. Here we explain the Lefschetz thimble method by using the example of simple one-dimensional integrals. Consider a complex action of a real variable $S(x)$. Next, extend the real axis to the complex plane $\mathbb{R} \to \mathbb{C}$, i.e. $x \to z = x + iy$. Given the stationary points $z_\sigma$ of $S(z)$

$$\left.\frac{\partial S}{\partial z}\right|_{z=z_\sigma} = 0 \,, \tag{1}$$

one can define a real path in the complex plane $D_\sigma \subset \mathbb{C}$ as the solution of the steepest descent equation ending at $z_\sigma$

$$\frac{\partial}{\partial \tau}z = -\overline{\frac{\partial S}{\partial z}} \,, \tag{2}$$

this path is called a Lefschetz thimble. The action has constant imaginary part along the thimble. The integral can be decomposed into integrals over all $D_\sigma$

$$Z = \int_D dz\, e^{-S} = \sum_\sigma n_\sigma e^{-i\text{Im}[S(z_\sigma)]} \int_{D_\sigma} dz\, e^{-\text{Re}[S(z)]} \equiv \sum_\sigma n_\sigma e^{-i\text{Im}[S(z_\sigma)]} Z_\sigma \,, \tag{3}$$

where $D \subset \mathbb{R}$ is the original real domain and $n_\sigma$ is the intersection number of the steepest ascent path (unstable thimble) with the original domain $D$. One can now formulate Monte Carlo algorithms based on (3), see e.g. [17, 23, 32]. Observables are then computed in the usual way

$$\begin{aligned}
\langle \mathcal{O} \rangle &= \frac{1}{Z} \sum_\sigma n_\sigma e^{-i\text{Im}[S(z_\sigma)]} \int_{D_\sigma} dz\, \mathcal{O} e^{-\text{Re}[S(z)]} \\
&= \frac{1}{Z} \sum_\sigma n_\sigma e^{-i\text{Im}[S(z_\sigma)]} Z_\sigma \langle \mathcal{O} \rangle_\sigma \\
&= \frac{\sum_\sigma n_\sigma e^{-i\text{Im}S(z_\sigma)} Z_\sigma \langle \mathcal{O} \rangle_\sigma}{\sum_\sigma n_\sigma e^{-i\text{Im}S(z_\sigma)} Z_\sigma} \,, 
\end{aligned} \tag{4}$$
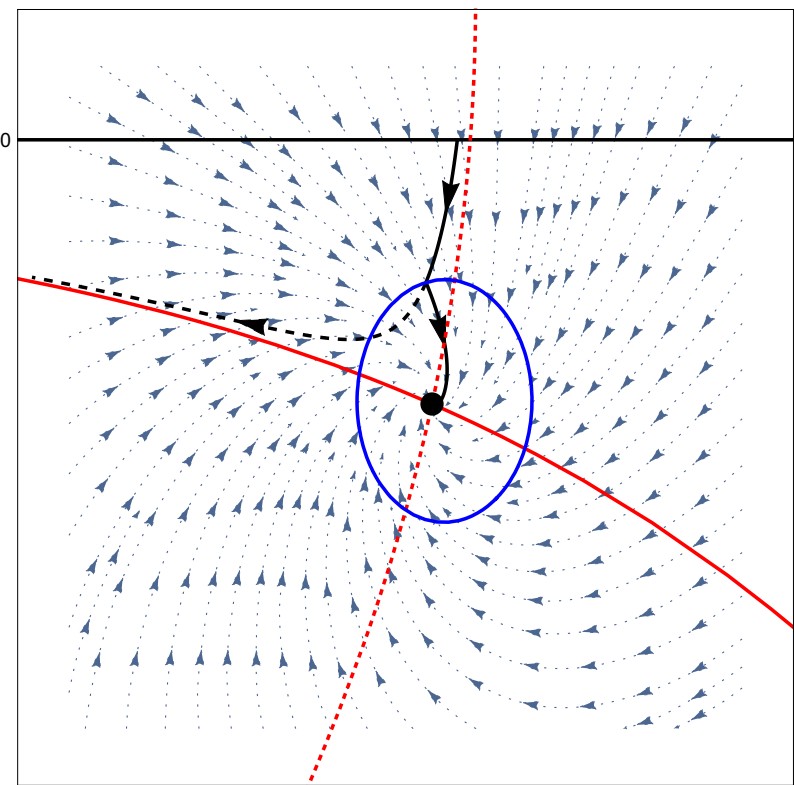

Figure 1: Visualization of the fixed point search via an axis scan. Red are the thimble (solid) and the anti-thimble (dashed). The steepest ascent equation (5) is solved using a starting point on the real axis close to the anti-thimble, once it is close to the fixed point (i.e. the derivative of the action is smaller than some value $\delta$, visualized by the blue circle), the flow is switched according to equation (8), and will end in the fixed point (solid black arrows). The steepest ascent without switching close to the fixed point will asymptotically approach the thimble (black dashed arrow).

with the only difference, that it has to be computed on every thimble. There are two practical problems with this approach:

1. Finding all contributing thimbles can be a challenging task.

2. For the case of multiple contributing thimbles there is no simple way so far to access the relative weights, i.e. the ratio $Z_{\sigma_i}/Z_{\sigma_j}$.

Both are challenging problems without a general solution so far. An approach to the first problem is the holomorphic gradient flow, which approximates the thimble structure, by simulating on a deformation of the original domain close to thimbles [23]. In [21] the second problem regarding relative weights has been approached by using known results in some parameter regions.

In the following we propose general solutions to both of those problems which do not rely on approximations. In this work we demonstrate our solutions by means of simple models.

# 3 Finding thimbles

In this section we propose two algorithms which can be used to systematically find contributing thimbles. This is put to work in simple one-dimensional integrals. Generalizations to higher dimensions might be expensive. The first algorithm scans the real axis in search of intersecting anti-thimbles, while the second algorithm projects points in the complex plane onto thimbles, in order to determine a numerical parametrization of the thimbles. Both algorithms also apply in higher dimensions, however the numerical costs may rise exponentially with the number of lattice points. This is currently investigated for gauge theory in [33].

## 3.1 Axis scan

Since the only contributing thimbles are those with non-zero intersection number of the anti-thimble with the original manifold, one can find all contributing fixed points by scanning the manifold for such intersections. This can be a challenging problem in higher dimensional theories, however importance sampling by Monte Carlo methods in parameter regions without a sign problem or in the phase quenched theory might give good starting points for such searches. In the following we describe the searching algorithm for the case of simple integrals, i.e. the original manifold is an interval $[a, b] \in \mathbb{R}$. The algorithm is the following [34],

1. Choose a starting point on the real axis.

2. Solve the steepest ascent equation

$$\frac{\partial z}{\partial \tau} = \overline{\frac{\partial S}{\partial z}} \Big/ \left| \overline{\frac{\partial S}{\partial z}} \right| , \tag{5}$$

   using the starting point as an initial condition.

3. If the derivative of the action becomes small

$$\left| \frac{\partial S}{\partial z} \right| < \delta , \tag{6}$$

   the flow is close to a fixed point of the action.

4. Depending on the structure of the fixed point, one can now reach it by looking at the Langevin flow (LF)

$$\dot{z} = -\frac{\partial S}{\partial z} , \tag{7}$$

   and changing the sign according to the following prescription

$$\dot{z} = \begin{cases} -\frac{\partial S}{\partial z} & \text{FP attractive under LF} \\ +\frac{\partial S}{\partial z} & \text{FP repulsive under LF} \\ \pm e^{i\pi/2} \frac{\partial S}{\partial z} & \text{FP circular under LF} \end{cases} . \tag{8}$$

   All those cases have to be tested, and one of them will end in the fixed point.

This algorithm is visualized in Fig. 1.

Once the fixed points are known, the numerical parametrization of the thimbles can be computed. In the case of one dimensional integrals, this boils down to solving one dimensional differential equations. We do so by solving the normalized steepest descent equation

$$\frac{\partial z}{\partial \tau} = -\overline{\frac{\partial S}{\partial z}} \Big/ \left| \overline{\frac{\partial S}{\partial z}} \right| , \tag{9}$$

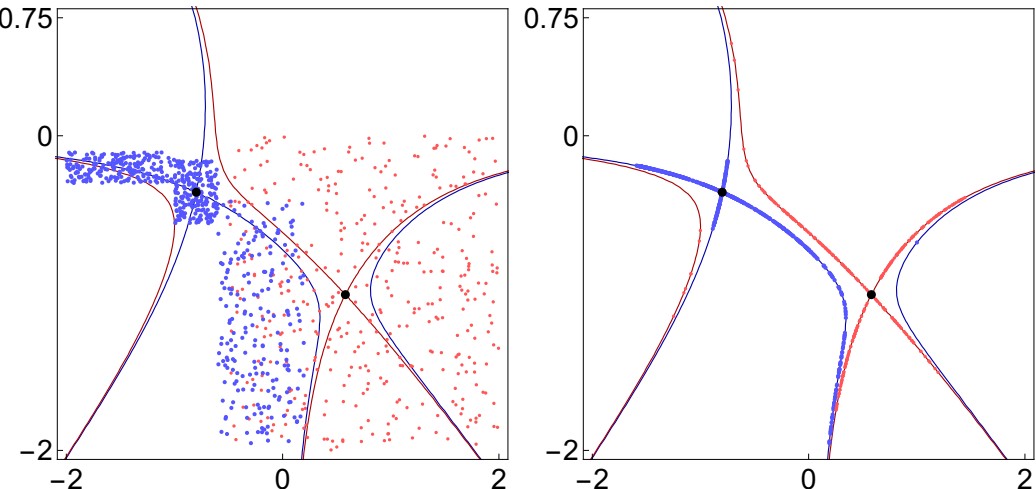

Figure 2: Complex plane with the solutions to the gradient equations (11) for a given grid of initial conditions for the left thimble (blue) and the right thimble (red). In the left plot we show a possible choice of initial conditions for the gradient equations. The right plot depicts the resulting distribution of points lying on the thimble and anti-thimble. Moreover other curves not passing through the fixed point on which the imaginary part of the action is constant are hit by the algorithm.

with opposite sign starting close to the fixed point [31]. The reason for the normalization with the absolute value will become clear later, however it also helps with numerical stability when solving the steepest descent equation. Note that this normalization is simply a rescaling of the flow parameter $\tau$.

## 3.2 Thimble cooling

In this section we propose a straight-forward algorithm to find parametrizations of all thimbles and anti-thimbles for a given action where the only required information is the knowledge about the fixed points. We show results for the $z^4$ model with parameters $\sigma = 1, \lambda = 1, h = 1+i$. For a definition of this model see Sec. 5.1. The idea is to minimize the distance of any set of points in the complex plane to curves on which the imaginary part of the action is constant. This leads to the following definition.

$$M_\sigma(x,y) = |\text{Im}S(x + iy) - \text{Im}S(z_\sigma)|^2, \tag{10}$$

where $\sigma$ labels the stationary points. We call $M_\sigma(x,y)$ the cooling function. The numerical minimization procedure is facilitated by the following gradient equations

$$\dot{x} = -\frac{\partial M_\sigma(x,y)}{\partial x},$$
$$\dot{y} = -\frac{\partial M_\sigma(x,y)}{\partial y}. \tag{11}$$

By construction the gradient equations orthogonally project a given point on a curve with constant imaginary part of the action. As initial conditions for (11) we use a grid of points in the complex plane. This is shown in the left plot in Fig. 2. The choice of a random grid is arbitrary. We could have also chosen a regular grid. In the right plot of Fig. 2 we show the result of the solution to the gradient equations. The procedure works well for a large box of initial conditions (red) as shown for the right fixed point (black). The resulting set of

points lies on the thimble, the anti-thimble and an additional curve on the left without physical relevance. Alternatively, we can start with a small rectangle around the fixed point, see the blue points in the left plot. From the resulting points flowed to thimble and anti-thimble we can choose the next set of initial conditions along e.g. the thimble and repeat the procedure iteratively. Note that also from looking at the flow lines of (11) we can determine suitable areas for initial conditions.

This method provides a useful tool to find parametrizations to the thimbles (and anti-thimbles) by interpolating the flowed set of points. Moreover, by knowing each anti-thimble we can map out if it intersects with the original integration manifold thus enabling us to determine whether and how much the corresponding thimble contributes. In particular, the method could be applied in higher dimensional theories where the minimization procedure is combined with importance sampling around the fixed point. From there thimbles and anti-thimbles can be successively parametrized as illustrated in blue in Fig. 2.

Thimble cooling has the potential advantage over the procedure in Sec. 3.1 that one does not have to solve the holomorphic gradient flow in many directions but one directly flows to the (anti-)thimbles. The cooling method put forward here can also be used to reduce numerical discretizations artifacts (see Sec. 5) in the thimble parametrization.

We remark, that a generalization of thimble cooling to higher dimensional integrals may in general prove difficult due to the dimensionality of the hyper-surface parametrized by $\text{Im}[S(z)] = \text{const}$. In App. C.3 we propose a combination of Lefschetz thimble and complex Langevin, which samples around all thimbles. This can also be used as a starting point for thimble cooling.

# 4 Monte Carlo simulations on Lefschetz thimbles

The previous section dealt with finding parametrizations for the thimbles. In this section we propose an algorithm for simulating on thimbles provided its parametrization is known. We also show how to compute the ratio of partition functions from within Monte Carlo simulations.

## 4.1 Reweighting on thimbles

Once the parametrization of the thimble is known, we can simply rewrite the partition function $Z_\sigma$ on the thimble as

$$\int_{D_\sigma} dz\, e^{-\text{Re}[S(z)]} = \int_a^b d\tau\, e^{-\text{Re}[S_\sigma(\tau)]} J_\sigma(z(\tau)), \tag{12}$$

where $S_\sigma(\tau) = S(z(\tau))$ is the action evaluated on the thimble $D_\sigma$. We have also rewritten the integral to run over the flow parameter in (9) and introduced integral boundaries, which are defined by the domain of $\tau$. This introduces the complex Jacobian $J_\sigma = \partial z/\partial \tau$ on $D_\sigma$. For the right hand side of (12) we can apply a real Langevin simulation or Monte Carlo sampling along the thimble. The Jacobian is dealt with via reweighting,

$$\langle \mathcal{O} \rangle = \frac{\langle \mathcal{O} J_\sigma \rangle}{\langle J_\sigma \rangle}. \tag{13}$$

Monte Carlo sampling now produces samples according to the distribution

$$p_i(\tau_i) = e^{-\text{Re}[S(\tau_i)]}. \tag{14}$$

So far we have dealt with a single thimble. The reweighting equation (13) for multiple thimbles becomes

$$\langle \mathcal{O} \rangle = \frac{\sum_\sigma n_\sigma e^{-i\text{Im}[S(z_\sigma)]} Z_\sigma^r \langle \mathcal{O} J_\sigma \rangle_\sigma^r}{\sum_\sigma n_\sigma e^{-i\text{Im}[S(z_\sigma)]} Z_\sigma^r \langle J_\sigma \rangle_\sigma^r} \,, \tag{15}$$

where we have defined

$$\langle \mathcal{O} \rangle_\sigma^r = \frac{1}{Z_\sigma^r} \int_a^b d\tau \, e^{-\text{Re}[S_\sigma(\tau)]} \mathcal{O} \,, \tag{16}$$

with

$$Z_\sigma^r = \int_a^b d\tau \, e^{-\text{Re}[S_\sigma(\tau)]} \,. \tag{17}$$

Note that in (15) the thimbles are weighted with their partition functions which have to be determined within the simulation.

## 4.2 Computing the partition function weights

Now that we have a simple algorithm for computing observables on the thimbles, we can proceed to the problem of how to compute the weights. With the above definition of $S_\sigma$ and considering only two thimbles for simplicity, we look at the ratio of their partition functions, i.e. we choose one thimble as a "master" thimble and divide the numerator and denominator of (15) by its partition function. The following identity states the ratio of partition functions

$$\frac{Z_1^r}{Z_2^r} = \left\langle e^{\text{Re}[S_2 - S_1]} \right\rangle_2^r \,, \tag{18}$$

provided (i) the integrals over the thimbles have the same boundaries and (ii) the flow parameters $\tau$ on both thimbles can be identified – if the latter does not hold, an additional Jacobian must be taken into account. For a derivation of (18) see App. A. (i) can be enforced by using suitable variable transformations, see App. B, while (ii) is guaranteed by normalizing the steepest descent equations, as we did in (5). Hence, it is possible to compute the ratio during the Monte Carlo simulation, which is necessary in higher dimensional integrals, e.g. in field theories.

# 5 Applications

We investigate different models with varying complexity to test our algorithms. First we look at a model with only one contributing thimble, which is a good test case for setting up the Monte Carlo simulation. Next we address a model with two contributing thimbles, whose flow parameters run over the same interval $\tau \in [-\infty, \infty]$. Finally we investigate the U(1) one link model, which is a model for a simple gauge theory with fermions and has thimbles that end in poles. This model is quite general in the sense that it contains all features that are to be expected in more complicated cases such as field theories.

## 5.1 One-site $z^4$ model

The one-site $z^4$ model generically consists of three thimbles, which end in different asymptotic regions at infinity. This structure makes the model a rather simple test case for the algorithms we propose. The model is given by the action

$$S(z) = \frac{\sigma}{2} z^2 + \frac{\lambda}{4} z^4 + hz \,, \tag{19}$$

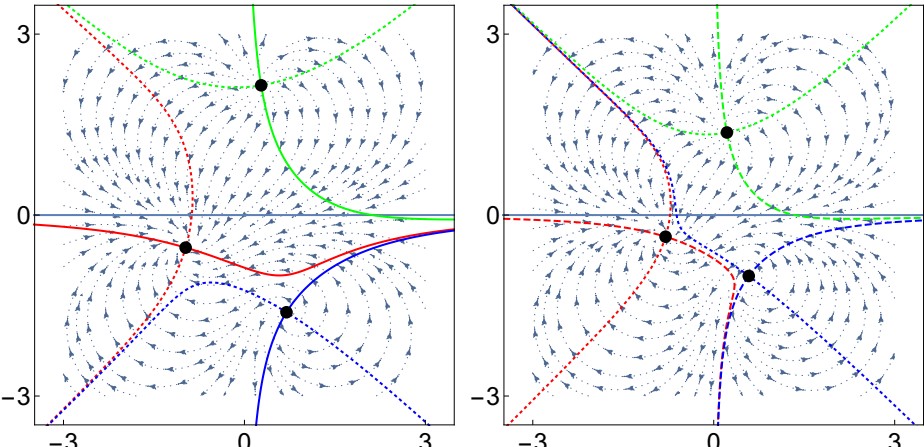

Figure 3: Complex plane with drift and thimble structure of the $z^4$ one-site model with one (left) and two (right) contributing thimbles.

Table 1: Numerical results and exact values of observables with statistical errors. Possible deviations are caused by numerical discretization errors.

| $\mathcal{O}$ | numerical | exact |
|---|---|---|
| $z^4$–1 thimble | | |
| $\mathrm{Re}z^2$ | 0.73922(6) | 0.73922 |
| $\mathrm{Im}z^2$ | 0.63006(4) | 0.630089 |
| $z^4$–2 thimbles | | |
| $\mathrm{Re}z^2$ | 0.509299(5) | 0.509297 |
| $\mathrm{Im}z^2$ | 0.305819(3) | 0.305815 |
| $Z_2/Z_1\vert_{T_1}$ | 0.2253778(4) | 0.2253779 |
| $Z_1/Z_2\vert_{T_2}$ | 4.436(12) | 4.437 |

for more details see e.g. [31]. We can choose the models parameters such that there are one or two contributing thimbles, i.e. with $n_\sigma \neq 0$:

1. For $\sigma = 1$, $\lambda = 1/3$ and $h = 1 + i$ there is only one contributing thimble.

2. For $\sigma = 1$, $\lambda = 1$ and $h = 1 + i$ there are two contributing thimbles.

Both cases are shown in Fig. 3.

The distributions $\exp(-\mathrm{Re}(S))$ on both thimbles are shown in Fig. 4, where they have been mapped onto the interval $[0, 1]$, see App. B. Note that this is not necessary for the simulation in this model, since the original domains already overlap. There we see that both distributions fall off exponentially which guarantees numerically stable simulations. Numerical results for the observable $\langle z^2 \rangle$ as well as the ratio of partition functions from (18) are given in Table 1, those simulations have been performed without employing the variable transformation for simplicity. For the simulations we have collected $(10^{10})$ data points. Errors have been estimated via a standard Jackknife analysis.

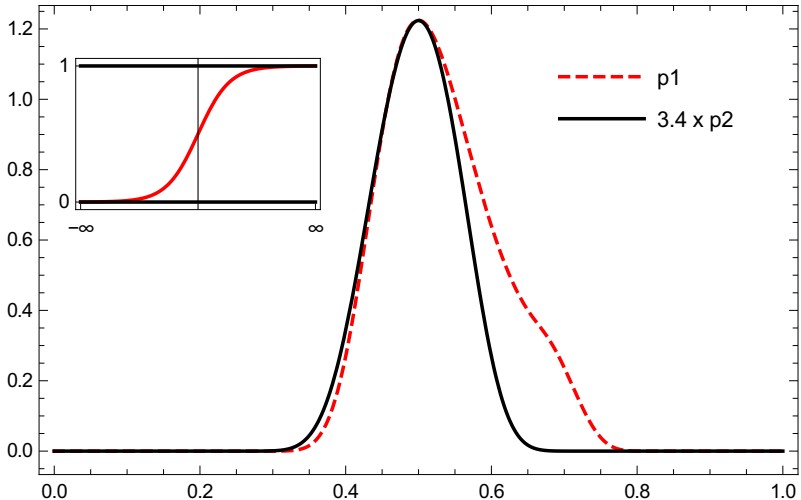

Figure 4: Boltzmann factor $\exp(-\mathrm{Re}(S))$ vs. flow parameter $\tau$ on the contributing thimbles in the $z^4$ model. The integration ranges have been mapped to $[0,1]$ and the transformation is shown in the inlay, for details see App. B. Here we choose the free parameter in the transformation (25) to be $\xi = 0.25$

## 5.2 U(1) one link model

After having demonstrated the viability of our methods in a simple model, we look at a more difficult model, namely the U(1) one link model with a finite chemical potential $\mu$. In addition to having multiple contributing thimbles, this model has thimbles that end in poles at finite values of the flow time making it necessary to transform the integrals such that the the distributions on the thimbles overlap. This model is a suitable testbed for our methods, since it contains general features that will also be present in more realistic theories. The model's action reads

$$S(x) = -\beta \cos(x) - \log(1 + \kappa \cos(z - i\mu)),\tag{20}$$

where $\kappa = 2$, $\beta = 1$ and $\mu = 2$. Its thimble structure is shown in Fig. 5.

This model was studied by means of complex Langevin in [14]. Its thimble structure has been studied in [26, 31]. It has three different contributing thimbles, of which two are connected by symmetry, see Fig. 5. Looking at the Langevin drift in the complex plane, one can see that there are two poles, in which all three thimbles end. At those poles the drift diverges. The distributions after mapping the integration on the finite interval $[0,1]$ according to App. B are given in Fig. 6. Note that in this model, only periodic observables which are analytic on U(1) make sense, hence we study the analogue of the Polyakov loop, its inverse, the plaquette and the density, which are given analytically in [14]

$$\langle U \rangle = \left\langle e^{ix} \right\rangle,$$

$$\left\langle U^{-1} \right\rangle = \left\langle e^{-ix} \right\rangle,$$

$$\langle P \rangle = \langle \cos(x) \rangle,$$

$$\langle n \rangle = \left\langle \frac{i\kappa \sin(x - i\mu)}{1 + \kappa \cos(x - i\mu)} \right\rangle.\tag{21}$$

Simulation results are given in Table 2. For the simulations we have collected $(10^9)$ measurements for the U(1) model. Again, errors have been estimated via a standard Jackknife analysis.

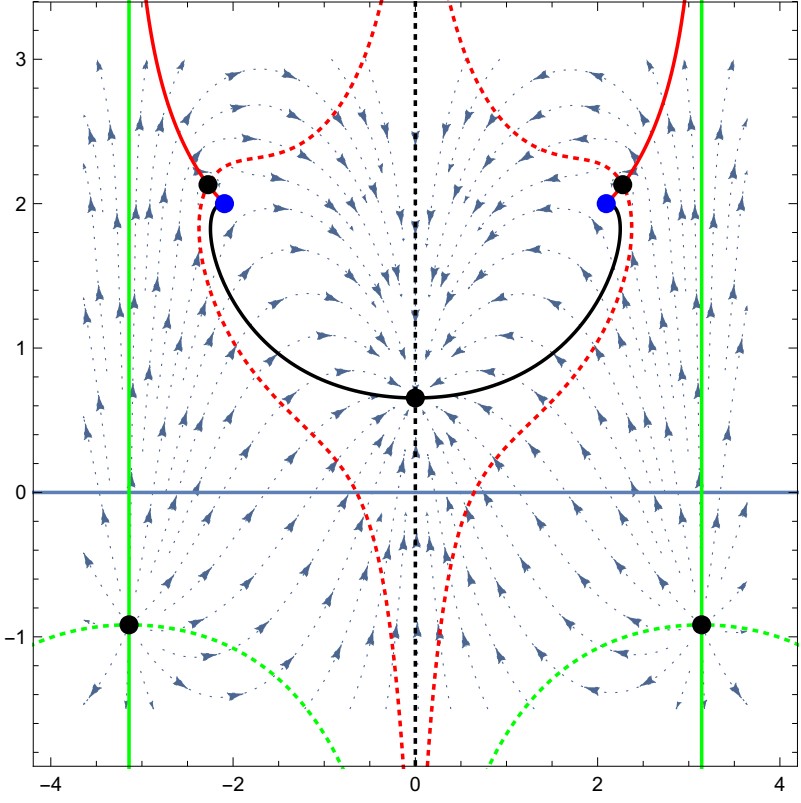

Figure 5: Complex plane with thimble structure and drift of the U(1) one link model. Note that due to periodicity the green thimbles (full vertical lines) which are not contributing are actually the same.

Table 2: Numerical results and exact values of observables for the U(1) model with statistical errors. Note that the imaginary parts for the observables are all consistent with zero within the statistical error. Possible deviations are caused by numerical discretization errors, see main text for a detailed discussion.

| $\mathcal{O}$ | numerical | exact |
|:---:|:---:|:---:|
| $\mathrm{Re}\langle U \rangle$ | 0.315217(3) | 0.315219 |
| $\mathrm{Re}\langle U^{-1} \rangle$ | 1.800941(3) | 1.800939 |
| $\mathrm{Re}\langle P \rangle$ | 1.058079(3) | 1.058079 |
| $\mathrm{Re}\langle n \rangle$ | 0.742861(1) | 0.742860 |
| $Z_2/Z_1\vert_{T_1} \times 10^{-3}$ | 2.99378(3) | 2.99382 |
| $Z_1/Z_2\vert_{T_2} \times 10^{4}$ | 3.34032(4) | 3.34022 |
| $Z_2/Z_3\vert_{T_3} \times 10^{-3}$ | 2.99377(3) | 2.99382 |
| $Z_3/Z_2\vert_{T_2} \times 10^{4}$ | 3.34026(9) | 3.34022 |

Our simulation results agree with the exact results from [31]. In Table 2 we only provide statistical errors. Systematic errors arise from numerical discretization artifacts along the

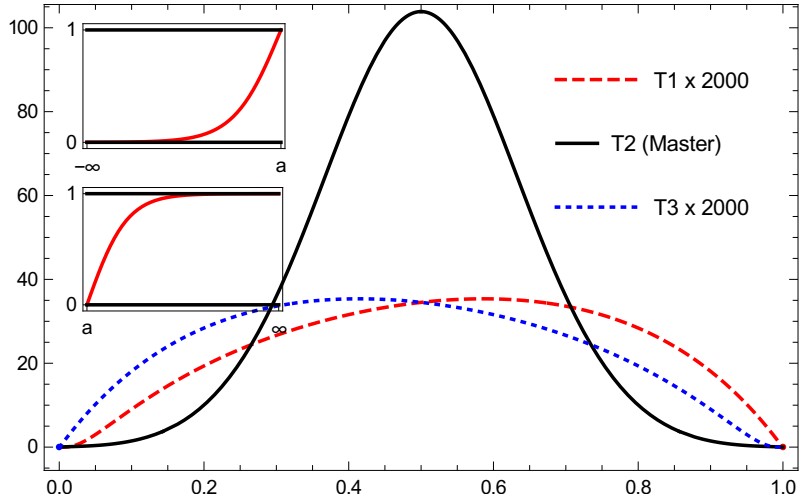

Figure 6: Boltzmann factor $\exp(-\mathrm{Re}(S))$ vs. flow parameter $\tau$ on the contributing thimbles in the U(1) one link model, where thimbles end in poles. Here it is necessary to transform integration ranges to maximize overlap. We chose to map all integration ranges to $[0,1]$, which can be done by a linear transformation for a thimble on $[a,b]$ or appropriate hyperbolic tangent transformations for intervals $[-\infty, a]$ or $[a, \infty]$, see inlays and (23) and (24) for the latter two transformations (we chose the free parameter of the transformation as $\xi = 1.5$). For details see App. B. This choice is not unique. Note that the curves have been rescaled for better visibility.

thimble. In the simple cases at hand the latter can be quantified by comparing the exact solution with the result from integrating along the discretization obtained from the gradient flow (11). In the case of the observables given in (21) the deviation is of order $10^{-6}$. Therefore the systematic error is comparable to the statistical error. The ratio of partition functions seems to be particularly sensitive to this effect. However, by taking into account the statistical and the expected systematic error all quantities agree with the exact result within the error.

# 6 Conclusions and outlook

We propose a new method based on Lefschetz thimbles for solving theories with a sign problem. This method works with two steps: First we find all contributing fixed points by scanning the original manifold for intersecting anti-thimbles. We then project a grid of points in the complex plane onto the thimbles – which requires the knowledge of the fixed points – to obtain a numerical parametrization. Second we simulate on these parametrizations and determine the relative weights within the simulation by means of a reweighting procedure. This reweighting can be tuned such that there is no overlap problem. We remark that within this method finding numerical parametrizations of the thimbles may be costly in higher dimensions. The reweighting procedure on the other hand straightforwardly generalizes to field theories and can be combined with other simulation algorithms for thimbles.

The above method has emerged from discussions and investigations of the idea of simulating a complex Langevin evolution, that is directed to the Lefschetz thimbles. This procedure of a Lefschetz-cooled Langevin update has been partially, but not fully, successful. In our opinion such a combined approach still has its potential, more details can be found in App. C.

Our proposal for simulating on Lefschetz thimbles has been illustrated on a one-site $z^4$

model in Sec. 5.1. It is successfully tested in a U(1) one-link model at finite density. The results are discussed in detail in Sec. 5.2.

Interesting future applications are field theories, e.g. the Schwinger model and higher-dimensional gauge theories, see [33].

# Acknowledgements

We thank K. Fukushima, C. Schmidt, A. Rothkopf, F. Ziesché, the Heidelberg Lattice group and the CLE collaboration for discussions and work on related subjects. This work is supported by EMMI, the BMBF grant 05P12VHCTG, and is part of and supported by the DFG Collaborative Research Centre "SFB 1225 (ISOQUANT)". I.-O. Stamatescu and M. Scherzer acknowledge financial support from DFG under STA 283/16-2. F.P.G. Ziegler is supported by the FAIR OCD project.

# A  Partition function weights

Here we give the derivation of (18).

$$\frac{Z_1^r}{Z_2^r} = \frac{\int_a^b d\tau \, e^{-\mathrm{Re}[S_1(\tau)]}}{\int_a^b d\tau \, e^{-\mathrm{Re}[S_2(\tau)]}} = \frac{\int_a^b d\tau \, e^{-\mathrm{Re}[S_1(\tau)+S_2(\tau)-S_2(\tau)]}}{\int_a^b d\tau \, e^{-\mathrm{Re}[S_2(\tau)]}}$$

$$= \frac{\int_a^b d\tau \, e^{-\mathrm{Re}[S_2(\tau)]} e^{\mathrm{Re}[S_2(\tau)-S_1(\tau)]}}{\int_a^b d\tau \, e^{-\mathrm{Re}[S_2(\tau)]}} = \left\langle e^{\mathrm{Re}[S_2(\tau)-S_1(\tau)]} \right\rangle_2^r . \tag{22}$$

This derivation of the case given in (22) requires two presuppositions

- The flow parameters on both thimbles can be identified. We normalize the steepest descent equation in order to automatically fulfill this requirement. Note that for cases where different parametrizations occur, instead of one flow parameter $\tau$, there will be $\tau_1$ and $\tau_2$ and the derivative $d\tau_1/d\tau_2$ should be taken into account. However, for practical purposes it should be possible to normalize the steepest descent equations such that $\tau_1 = \tau_2$.

- The integration boundaries are the same. This can be enforced easily by variable changes in the integral, see App. B. Note that if the integration boundaries are the same from the beginning as for the case of (22), then there will be no overlap problem, since the fixed points give the main contribution on the thimbles and we chose our parametrization such that all fixed points correspond to $\tau = 0$. Hence the peaks of the distributions are at the same point.

# B  Mapping integration ranges

If different thimbles have different parameter ranges, one has to map all of them to the same interval, here we choose the interval $[0, 1]$. In case of an integral in the range $[a, b]$, a simple linear shift is enough. For an integral over $\tau \in [-\infty, a]$, one possible transformation is

$$x \to x' = 1 + \tanh\left(\xi(x - a)\right), \tag{23}$$

conversely for $\tau \in [a, \infty]$, the analogue is

$$x \to x' = \tanh(\xi(x-a)),\tag{24}$$

and for $\tau \in [-\infty, \infty]$ the mapping becomes

$$x \to x' = \frac{1 + \tanh(\xi x)}{2},\tag{25}$$

where the parameter $\xi$ can be chosen such that the overlap of the distributions in (22) is maximal and hence the overlap problem becomes small. The Jacobian of the transformation can then be absorbed in the action for the Monte Carlo simulation. We chose this transformation in the case of the U(1) one link model, where we chose $\xi = 1.5$. We only choose such transformations that have sufficiently fast falloff at the boundaries such that those regions are suppressed exponentially.

## C  Combining the Complex Langevin and Lefschetz thimble methods

As there are many complicated steps for simulations on thimbles, it is desirable to find simpler alternatives, which at best can be applied blindly. One natural idea [32, 34–37] is to combine the complex Langevin evolution with the Lefschetz thimbles. A combination of both equations is only consistent after an appropriate coordinate transformation. The latter can adaptively be generated during the combined Langevin and gradient flows. Due to its similarity to standard cooling algorithms as well as the gauge cooling we call this process *Lefschetz cooling*.

Despite its full success described below in particular for simple Gaußian models it is only partially successful in more complicated models, notably already the $z^4$ model. While the method is not fully successful yet, in our opinion it is still a very interesting one to pursue. Its potential power is the self-adaptive *local* nature of the simulation steps. However, this also poses the biggest conceptual question: How does such a local procedure capture the global nature of the intersection numbers $n_\sigma$ in (3) correctly? Note that besides its formal importance this question could be practically less important as it seems: in most models under investigation so far we have $n_\sigma = 1$.

Below, we list some ideas putting Lefschetz cooling to work and discuss their viability and applications. While none of those proposed ideas so far have managed to give quantitative correct results for observables, they provide useful insight into possible realizations of the approach.

### C.1  Variable transformations

We aim to make the complex Langevin evolution compatible with constraints characterizing the Lefschetz thimbles by means of variable transformations. The latter have been investigated in combination with the complex Langevin evolution in a different context in [38]. There it was shown that the complex Langevin evolution including a transformation can give correct results while failing in the original formulation of the problem. Here we pursue the idea of having flow time-dependent variable transformations to transform the complex Langevin evolution towards the thimbles of the theory. This approach is natural in the sense that complex Langevin should already be able to sample the relevant fixed points [31], note that this is in a similar spirit as for the contraction algorithm [23]. By forcing the complex Langevin evolution close to thimbles the sign problem should be weakened and parameter regions that have been inaccessible so far may be reached. In the following discussion we consider again

one-dimensional integrals. One rather general ansatz for such variable transformation is the Möbius transformation

$$z(u) := \frac{au + b}{cu + d}. \tag{26}$$

This rather general ansatz has four $\tau$-dependent parameters that have to be determined during the simulation. This turns out to be a rather challenging task. We find that the transformation (26) seems to introduce repulsive structures destabilizing the evolution. Hence, we focus on a special Möbius transformation, namely a rotation

$$z(u) := ue^{i\theta}, \tag{27}$$

where $u$ takes the role of the (complex) field variable and $\theta$ is a $\tau$-dependent parameter. Consider a point in the complex plane sufficiently close to the thimble. Then, a rotation suffices to map this point even closer to or onto the thimble. This indicates that the transformation (27) is both necessary and sufficient for fulfilling the constraints mentioned above. For the remaining part of this paper we always refer to the rotation (27) when discussing variable transformations.

## C.2 Lefschetz cooling

Thimbles are curves passing through the fixed points and along them the imaginary part of the action is constant. This gives rise to various constraints which we impose onto the complex Langevin evolution by including the variable transformation (27).

Let $u \in \mathbb{C}$ and $\theta \in \mathbb{R}$. The transformed action reads

$$S_u := S(z(u)) - \log(z'(u)). \tag{28}$$

The procedure here is to be understood as a passive transformation, see the appendix in [38]. Hence, the complex Langevin equation in the transformed variables becomes

$$\partial_\tau u = -\frac{\partial S_u}{\partial u} + \eta, \tag{29}$$

where $\eta \in \mathbb{R}$.

In the following we investigate how the $\tau$-dependent transformation parameter $\theta$ evolves under the dynamics induced by different constraints.

### C.2.1 Lefschetz cooling the transformed thimbles

First, we formulate the additional constraint completely in the transformed theory [35], i.e. we demand

$$\mathrm{Im}(S_u) = \mathrm{const}. \tag{30}$$

This constrains the evolution close to the thimbles in the transformed theory. By taking the total $\tau$ derivative of this equation we obtain

$$\mathrm{Im}\left(\frac{\partial S_u}{\partial u}\dot{u}\right) + \mathrm{Im}\left(\frac{\partial S_u}{\partial \theta}\right)\dot{\theta} = 0. \tag{31}$$

By inserting the Langevin evolution (29) we get the time evolution of the angle $\theta$

$$\frac{\partial \theta}{\partial \tau} = \frac{\mathrm{Im}\left(\left(\frac{\partial S_u}{\partial u}\right)^2\right)}{\mathrm{Im}\left(\frac{\partial S_u}{\partial \theta}\right)}. \tag{32}$$

### C.2.2 Lefschetz cooling the original thimbles

Alternatively, we may formulate the constraint in the original theory which yields

$$\text{Im}(S_z) = \text{const}. \tag{33}$$

Again taking the total derivative and inserting (29) the evolution for $\theta$ becomes

$$\frac{\partial \theta}{\partial \tau} = \frac{\text{Im}\left(\frac{\partial S_z}{\partial z}\frac{\partial z}{\partial u}\frac{\partial S_u}{\partial u}\right)}{\text{Im}\left(\frac{\partial S_z}{\partial z}\frac{\partial z}{\partial \theta}\right)}. \tag{34}$$

### C.2.3 Applicability

On both, the original and the transformed thimbles one can see that the denominator in the evolution equations for $\theta$ in (32) and (34) introduces poles.

We analyze these in the representation of the original variable $z$ and show the denominators of (32) and (34) for the $z^4$ model respectively in Fig. 7.

Those poles destabilize the numerical simulations and so far we have not found a way to resolve this. Our efforts for improvements include modifications to the noise term such that the poles are penalized, i.e. one can multiply the noise by the denominator of (32) or (34) respectively. However, this (i) leads to wrong expectation values and (ii) prohibits the Langevin evolution from jumping between the contributing thimbles. This prevents sampling of the correct thimble weights. In simple models such as the one-site Gaußian model

$$S(z) = \frac{\sigma}{2}z^2, \tag{35}$$

(ii) is not a problem, since the model has only one thimble and (i) can be resolved by a linear rescaling. This is however due to the low complexity and high symmetry of the model and does not generalize.

### C.3 Advanced Lefschetz cooling

Instead of explicitly demanding the thimble constraint to be fulfilled, it is possible to directly combine the complex Langevin evolution with the steepest descent equations.

$$\frac{\partial u}{\partial \tau} = -\frac{\partial S_u}{\partial u} + \eta$$

$$\frac{\partial u}{\partial \tau} = -\overline{\frac{\partial S_u}{\partial u}}. \tag{36}$$

Here the idea is that complex Langevin already takes into account all relevant fixed points and the steepest descent equation keeps the evolution close to the thimbles, alleviating the sign problem. Taking the difference of the previous two equations implies that the imaginary part of $u$ remains constant which reduces the evolution in $u$ to real Langevin. Hence we have

$$0 = \frac{\partial S_u}{\partial u} - \overline{\frac{\partial S_u}{\partial u}} \Rightarrow \partial_\tau \text{Im}(u) = 0. \tag{37}$$

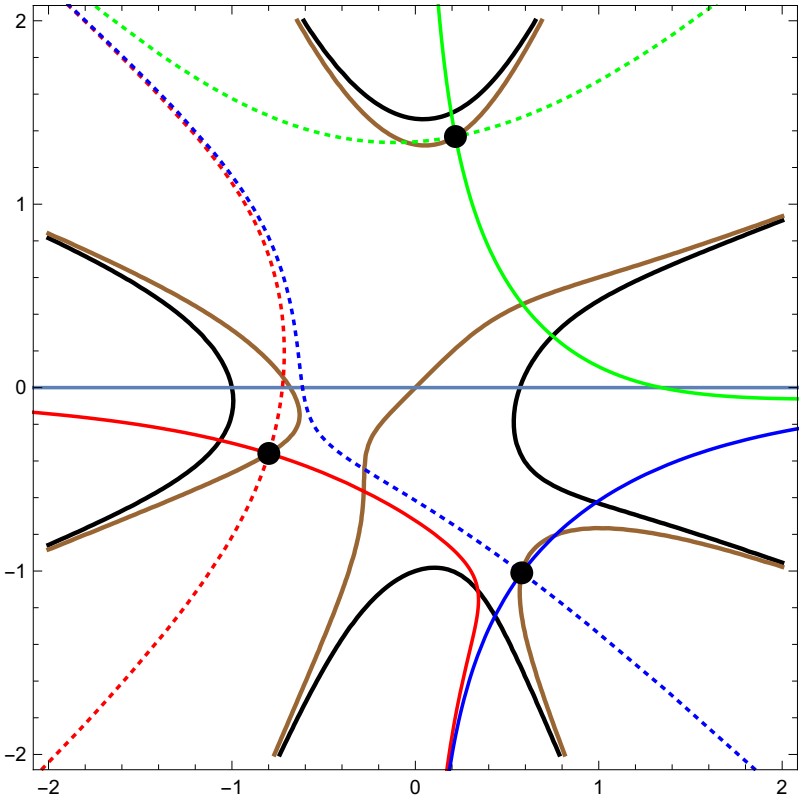

Figure 7: Pole and thimble structure for the $z^4$ model enforcing the two constraints (30) and (33), each represented in the original variable $z$. Red, green and blue solid (dashed) lines are the (anti-)thimbles and black points denote the fixed points. The brown solid lines are the poles from the constraint $\mathrm{Im}[S_z] = \mathrm{const}$, while the black solid lines represent the poles from the constraint $\mathrm{Im}[S_u] = \mathrm{const}$.

This leads to the following evolution equation for $u$ and conditions to the angle $\theta$. Thus we find by using the action as given in (28)

$$\partial_\tau \mathrm{Re}\, u = -\mathrm{Re}\, \frac{\partial S_u}{\partial u} + \eta$$

and

$$\partial_\tau \mathrm{Im}\, u = -\mathrm{Im}\, \frac{\partial S_u}{\partial u} = -\mathrm{Im}\left( \frac{\partial S}{\partial z} z' - \frac{z''}{z'} \right) = 0 \,. \tag{38}$$

Note that here we explicitly see, why a rotation should be sufficient. We illustrate this by means of the Gaußian model (35). It holds that $z' = e^{i\theta}$, $z'' = 0$ leading to the following expression for the constraint in (38)

$$\mathrm{Im}(\sigma z z') = \mathrm{Im}(\sigma e^{2i\theta} u) = 0 \,. \tag{39}$$

Here, $u \in \mathbb{R}$ and with $\sigma = \sigma_r e^{i\theta_\sigma}$ the solution to (39) becomes

$$\theta^* = -\frac{1}{2}\theta_\sigma \,. \tag{40}$$

This rotates the thimble in the original theory precisely onto the real axis of the transformed theory.

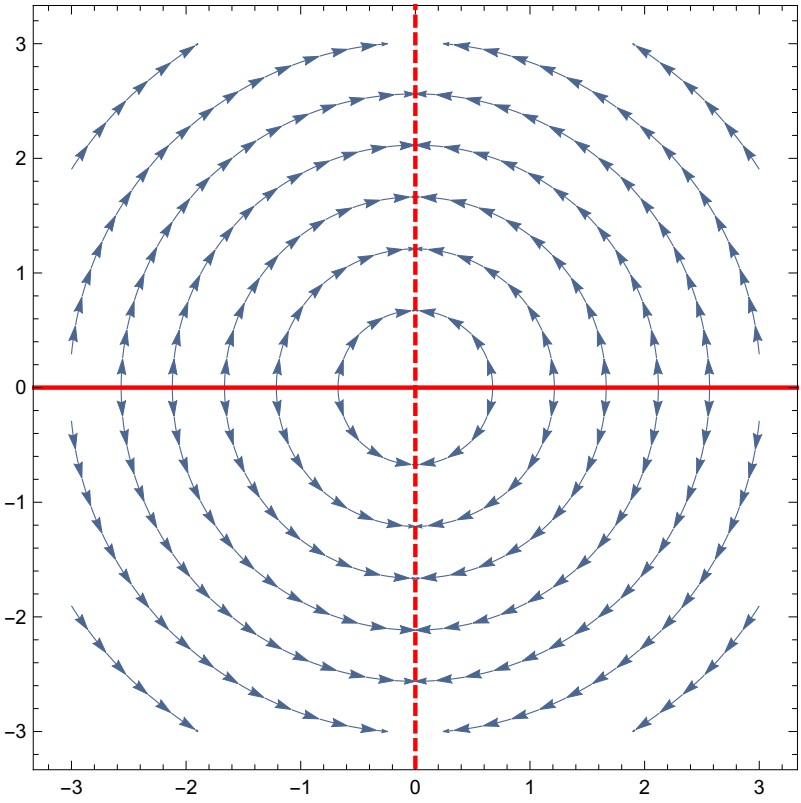

Figure 8: Visualization of the rotation of the complex plane for the Gaußian model with $\sigma = 1 + i$ according to (44) after rotating the thimble onto the real axis. The arrows point in the direction of rotation. Here, the transformed thimble is the real axis (solid red line, repulsive), while the anti-thimble is the imaginary axis (dashed red line, attractive).

Next, we investigate how $\theta$ changes with the flow induced by (37). Hence, we take the total $\tau$ derivative of (37), yielding

$$\frac{d}{d\tau} \text{Im}\left(\frac{\partial S_u}{\partial u}\right) = 0. \tag{41}$$

This leads to

$$\text{Im}\left(\frac{\partial^2 S_u}{\partial u^2} \dot{u}\right) + \text{Im}\left(\frac{\partial^2 S_u}{\partial u \partial \theta}\right)\dot{\theta} = 0, \tag{42}$$

from which we find

$$\dot{\theta} = \frac{\text{Im}\left(\dfrac{\partial^2 S_u}{\partial u^2} \dfrac{\partial S_u}{\partial u}\right)}{\text{Im}\left(\dfrac{\partial^2 S_u}{\partial u \partial \theta}\right)}, \tag{43}$$

upon inserting the drift term for $\dot{u}$. In this form, the dynamics are unstable. For instance in the Gaußian model, where the (anti-)thimble is a straight line this manifests itself by a diverging evolution along the anti-thimble to infinity. (40) rotates the thimble onto the real axis and transforms the anti-thimble into the imaginary axis.

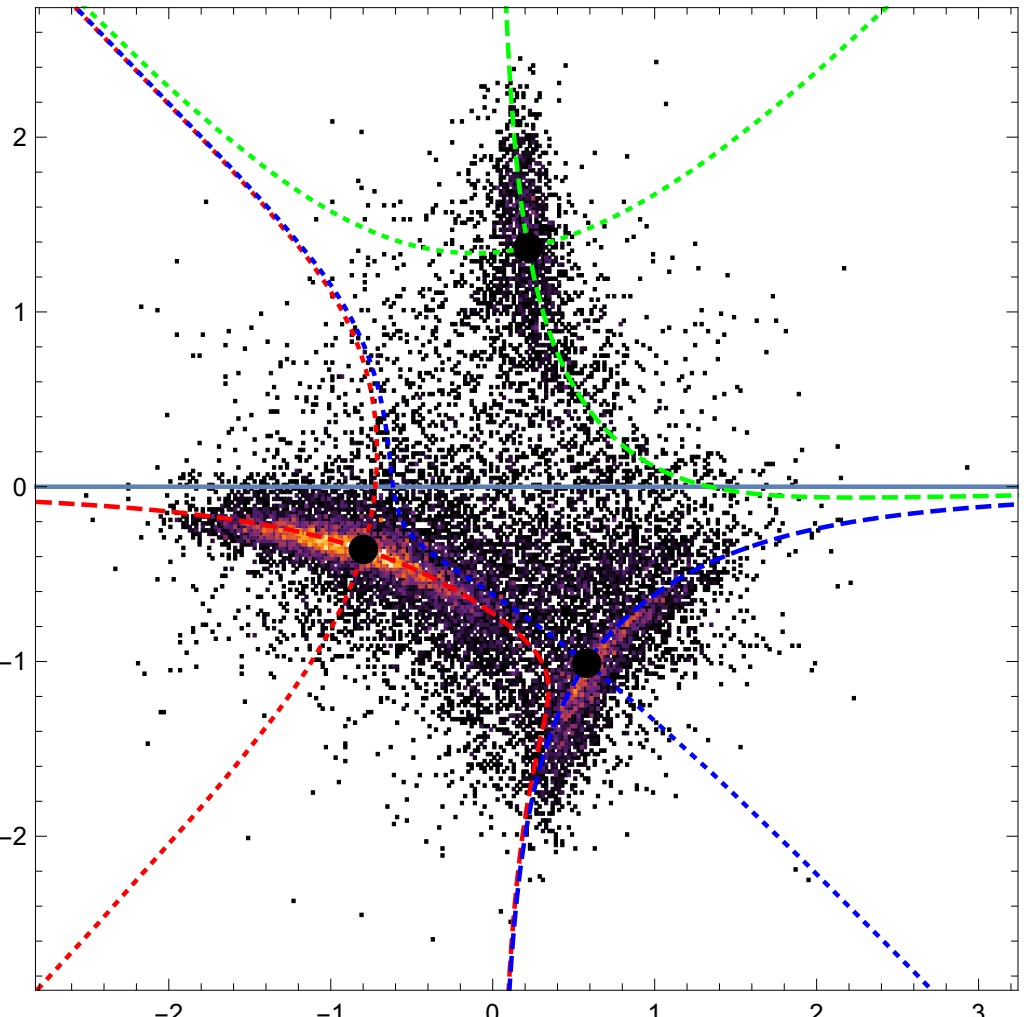

Figure 9: Scatter plot of the combined complex Langevin evolution and thimble constraint dynamics in $\theta$ with the reversed sign mapping, see (45) and the discussion in the text. The proposed method is being applied to the $z^4$ model and the evolution is represented in the original variable $z$. The scatter plot is color coded, where black corresponds to low density and yellow corresponds to high density. The mapping enforces the sampling of all relevant thimbles, as well as allowing for transitions between them and guarantees stability.

Inserting the action of the Gaußian model into (43) we find

$$\dot{\theta} = \frac{1}{\sigma_r}\sin(\theta_\sigma + 2\theta). \tag{44}$$

Examining the numerical solution to the previous equation we easily see that the thimble is repulsive whereas the anti-thimble is attractive, see Fig. 8 for an illustration represented in the transformed theory.

To render the thimbles attractive we consider again (42). Replacing $\dot{u}$ by the Langevin drift with a reversed sign $+\partial S_u/\partial u$ reverses the sign in (43) yielding

$$\dot{\theta} = -\frac{\mathrm{Im}\left(\dfrac{\partial^2 S_u}{\partial u^2}\dfrac{\partial S_u}{\partial u}\right)}{\mathrm{Im}\left(\dfrac{\partial^2 S_u}{\partial u \partial \theta}\right)}. \tag{45}$$

This also enables and enforces hopping between the thimbles and inverts the stability properties of the fixed points. Note that the evolution in $u$ is still governed by the complex Langevin equation 29).

Therefore, inserting the sign-reversed drift for $u$ in (43) can be understood in the sense of a mapping with the following properties: it guarantees that the evolution stays close to the thimbles, as well as allowing for transitions between different contributing thimbles. This approach yields the correct result for the Gaußian model. However, once the thimble structure becomes slightly more complicated, the values of observables are not computed correctly. This has been tested for different actions in [34, 36]. While the procedure samples all thimbles, it does not correctly take into account their relative weights. Fig. 9 shows a scatter plot of the $\tau$-evolution applied to the $z^4$ model. Clearly, this algorithm samples all thimbles. But the contributing ones are being sampled with a higher weight, see the yellow regions in the scatter plot.

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
