# Peer review of "Reweighting Lefschetz Thimbles"

_SciPost Physics, doi:SciPost Phys. 5, 044 (2018)_

## Round 1 · Referee Report · Anonymous · 2018-7-2

Strengths

1- genuine research
2- well structured and written

Weaknesses

none

Report

I propose to accept the paper ``Reweighting Lefschetz Thimbles'' for publication.

The Lefschetz Thimble method is a promising approach to address the finite baryon density sign problem. So far, however, the method has only been addressed in toy models with few degrees of freedom.

The paper is very well written and reflects the state of the art. Two of the main problems of the approach are addressed: finding all contributing thimbles (manifolds on which the imaginary part of the action is fixed), and determining the relative weights of the different thimbles if there is more than one.

In particular the idea of thimble cooling is original and may be applicable in higher-dimensional theories.

The residual sign problem on a single thimble introduced by the complex Jacobian $\partial{z}/\partial{\tau}$ may however be severe in some cases, particularly in higher dimensions.

One site models (such as the $z^4$ model) or one-link models are interesting enough to understand the thimble structure and check the proposed methods. It is impressive how well reweighting with respect to a selected master thimble works to compute the relative weights.

One question that could be addressed in the paper is: what is the expectation value of the Jacobian $<J_\sigma>_\sigma^r$ (due to deformation of parameter space), and as needed for Eq. 15, for the investigated models? This should indicate how strong the overlap of the Jacobian with the Boltzmann weight is.
The residual phase of the Jacobian (as e.g. discussed in ref. [23]) might pose problems in Monte Carlo.

My suspicion is that the overlap is good in low-dimensional models, but not for more realistic scenarios. If so, reweighting on Thimbles is again limited by the overlap problem.
In the conclusion, the authors however claim that the reweighting can be tuned (in general?) such that there is no overlap problem. Details would be helpful.

The discussion in appendix C, in particular the idea of combining the complex Langevin evolution and the steepest descent equation, is very interesting conceptually.

In summary, the paper contains genuine and relevant research and is very well structured. Hence I recommend it for publication.

Requested changes

none

  • validity: high
  • significance: high
  • originality: high
  • clarity: high
  • formatting: excellent
  • grammar: perfect

Author:  Felix Ziegler  on 2018-08-15  [id 306]

(in reply to Report 1 on 2018-07-02)
Category:
remark
answer to question

We thank the referee for her / his positive feedback to our manuscript.

We would like to reply to two of the referee's comments.

1.) Paragraph 7

One question that could be addressed in the paper is: what is the expectation value of the Jacobian $\langle J_{\sigma} \rangle^{r}_{\sigma}$ (due to deformation of parameter space), and as needed for Eq. 15, for the investigated models? This should indicate how strong the overlap of the Jacobian with the Boltzmann weight is. The residual phase of the Jacobian (as e.g. discussed in ref. [23]) might pose problems in Monte Carlo.

In the models considered in the manuscript at hand the calculation of the expectation value of the Jacobian $\langle J_{\sigma} \rangle^{r}{\sigma}$ does not suffer from a severe sign problem. We checked that the overlap of the Jacobian with the Boltzmann weight $e^{-\text{Re}(S)}$ is sufficiently large. Moreover, we quantitatively determined the expectation value of $\langle J \rangle^{r}_{\sigma}$ yielding a phase factor whose magnitude is close to one indicating that the sign problem is mild.

2.) Paragraph 8

My suspicion is that the overlap is good in low-dimensional models, but not for more realistic scenarios. If so, reweighting on Thimbles is again limited by the > overlap problem. In the conclusion, the authors however claim that the reweighting can be tuned (in general?) such that there is no overlap problem. Details would be helpful.

The overlap problem can be dealt with by suitable coordinate transformations as described in appendix B. In particular, this is accomplished by shifting the peaks of the different Boltzmann weights close together. In higher dimensions finding such coordinate transformations and hence optimizing the overlap might be difficult due to the effort of tuning an increasing number of parameters in the transformation.

---

## Round 1 · Referee Report · Anonymous · 2018-8-9

Strengths

1) studies a topical approach to an outstanding problem

Weaknesses

1) proposes a method to compute the simplest toy models possible which does not seem to generalize to realistic situations

Report

The paper proposes a method to address some difficulties found in some approaches to the "thimble method" used to deal with the sign problem in Monte Carlo calculations in field theory. It uses a toy model of the actual situation, namely , a bosonic one dimensional integral (not a field theory in one dimension !). That, by itself, is not a problem, even though the state of art in the field has moved to hundred of degrees of freedom in fermionic theories. The problem arises because one-dimensional integrals are fundamentally different from any other dimension, including two-dimensional integrals. The difference stems from the fact that the thimbles in 1D can be characterized by by the condition Im S(z)=0. The same condition on a space with 2N real dimensions (N complex dimensions) defines a manifold with 2N-1 dimensions, a much larger space than thimbles. That seems to preclude any attempt at generalizing the "thimble cooling" method advocated in the paper to any interesting situation. Also, the idea of mapping the location of the thimbles may work in one-dimensional integrals but would have a prohibitive cost in any field theory. Let alone the problem of finding the thimbles, just to store the results has a cost exponential on the spacetime volume. The paper should make very clear which aspects of their proposal can possibly be generalized to real field theories.

There are also problems in citation of the literature that seem to me drastic enough that requires correction. The problems of finding the thimbles and their corresponding statistical weights afflicts only one of the "algorithmic frameworks" (as the authors put it). This should be made clear not to misdirect the reader.

I would recommend the publication of the paper

Requested changes

1) The paper should make very clear which aspects of their proposal can possibly be generalized to real field theories, pointing out which steps would require exponentially growing resources.

2) The citations should describe the current state-of-art of the field. There are calculations performed in real field theories, not toy models. They should be cited. Even real time calculation have appeared. In addition, it should be pointed out how these calculations were possible, despite the difficulties with the "thimble approach" the authors list. Could it be the other methods can bypass them? This is important information to the reader and not discuss them detracts from the utility of the paper.

  • validity: ok
  • significance: low
  • originality: good
  • clarity: high
  • formatting: perfect
  • grammar: excellent

Author:  Felix Ziegler  on 2018-10-05  [id 327]

(in reply to Report 2 on 2018-08-09)
Category:
remark
correction

We thank the referee for her / his feedback to our manuscript.

We made a few changes to our manuscript, see resubmitted version (v2), in order to meet the requested changes by the referee.

1) The generalizability of methods proposed in our paper (in particular the thimble cooling method) has been addressed in the following sections:

  • towards the end of the introduction
  • at the beginning and at the end of Sec. III (thimble search algorithms)
  • in the conclusions

2) To our best knowledge the cited literature in v1 reflects the state-of-the-art in the field at the time of publication on arXiv (see also report by referee 1). Works dealing with field theories and even real-time computations have been already cited in the introduction. We have added references on the path optimization technique as well as recent preprints that were published after the submission of our manuscript.

---

## Round 2 · Author Response

This resubmission incorporates changes in answer to two referee reports.

---

## Round 2 · List of Changes

• updates on the list of references
  • detailed comments on the scalability of the methods

---

## Editorial Decision

published